# Welfare Assessment and Activities of Captive Elephants in Thailand

**DOI:** 10.3390/ani10060919

**Published:** 2020-05-26

**Authors:** Pakkanut Bansiddhi, Janine L. Brown, Chatchote Thitaram

**Affiliations:** 1Center of Elephant and Wildlife Research, Chiang Mai University, Chiang Mai 50100, Thailand; pakkanut.b@cmu.ac.th (P.B.); BrownJan@si.edu (J.L.B.); 2Department of Companion Animals and Wildlife Clinics, Faculty of Veterinary Medicine, Chiang Mai University, Chiang Mai 50100, Thailand; 3Center for Species Survival, Smithsonian Conservation Biology Institute, Front Royal, VA 22630, USA

**Keywords:** welfare, elephant, management, tourism, Thailand

## Abstract

**Simple Summary:**

In Thailand, captive elephants are used in tourism and involved in a variety of activities, such as feeding, bathing, riding, or just observation. The welfare of these elephants has been a topic of intense debate in recent years, resulting in divergent opinions on what are or are not appropriate uses. This paper summarizes the current status of captive elephants in Thailand, highlighting issues and challenges facing elephant tourism today. We review recent welfare studies conducted to understand how tourist activities affect welfare outcomes and provide recommendations for the future. Captive elephants in Thailand must be managed under human care with good practices aimed at meeting welfare needs to ensure healthy, sustainable populations. Our goal is to provide information to elephant facilities on best practices, and relay findings to governmental and tourist organizations in Thailand to ensure the welfare and sustainability of this important species.

**Abstract:**

Thailand is the epicenter of elephant tourism and visiting an elephant camp is a popular activity according to the Tourist Authority of Thailand. However, the welfare of these elephants has been questioned by animal activist groups, international tour operators, and the public. Conclusions that the vast majority of captive elephants are abused often are based on anecdotal evidence and not solid science. So, it is difficult to tease apart emotion, opinion, and fact with regard to what practices are good or bad for elephant welfare. The aim of this paper was to: (1) describe the unique status of captive elephants in Thailand and associated regulations, (2) summarize current issues and challenges facing elephant tourism, (3) review studies conducted on welfare of tourist elephants in Thailand, and (4) offer recommendations for how elephants can be properly cared for under captive conditions in tourist camps. We conclude there are many ways to manage these elephants, and that not all tourist activities are bad for welfare. However, it is essential they be managed in a way that meets physical, physiological and psychological needs, and that management decisions are based on objective data.

## 1. Overview of Captive Elephant Management and Tourism

The total number of elephants in Thailand is estimated to be 6783–7483, with 3783 being captive and 3000–3500 in the wild [1,2]. Elephants in Thailand have been under human care since the 13th century (Table 1) and used for a variety of purposes, e.g., war (like horses in western countries), religious and cultural ceremonies, and logging. After the logging ban in 1989, elephants have been used primarily for tourism. Beginning in the north, elephant camps have increased in number throughout the country [3,4]. Therefore, the use of captive elephants in tourism has been ongoing for almost five decades. Nowadays, Thailand is the epicenter of elephant tourism. The National Institute of Elephant Research and Health Service, Department of Livestock Development estimated there were 2700 elephants working in 250 tourist venues across the country in 2017 [5]. Activities in elephant camps vary and can include observation, feeding, bathing, walking, riding, and elephant shows [6]. Criticisms of venues that include tourist interactions with elephants have escalated over the past decade, with concerns that any type of physical contact between elephants and tourists should be stopped, requiring a re-evaluation of how these activities affect individual elephant welfare. 

## 2. Captive Elephant Laws and Regulations

Elephants in Thailand are classified as wild or captive, and each have their own set of regulations. Killing or capture of wild elephants has been prohibited since 1921 by the Wild Elephant Protection Act. Nowadays, Asian elephants are categorized as a protected animal under the Wildlife Protection and Protection Act of 1992, which has been updated regularly. Capturing, killing and injuring elephants are prohibited, and results in penalty. For nearly 100 years, then, captive populations have been sustained by captive breeding, with surprisingly little regulation of their management. Elephants are considered livestock, along with donkeys, oxen, buffalo, and other livestock (Draught Animal Act of 1939), and none are afforded animal welfare protection. Although killing or capture of wild elephants has been prohibited since 1921 by the Wild Elephant Protection Act, illegal capture and trade of wild elephants from Myanmar and other countries to support tourism in Thailand remained a significant concern [7]. So, in 2016, three organizations agreed to work cooperatively to register, census, microchip, and DNA fingerprint all captive elephants: the Department of Livestock Development (DLD), Ministry of Agriculture and Cooperatives; the Department of Provincial Administration, Ministry of Interior; and the Department of National Park, Wildlife and Plant Conservation, Ministry of Natural Resources (from Section 44 of the Interim Constitution enforced by the Thai government during September 2016–March 2017). One aim was to prevent the illegal capture of wild elephants for use in tourism, which has put pressure on elephant venues to maintain elephant numbers through captive breeding. In a survey of 33 camps in northern Thailand, elephants breed well in some, but not all camps, with 55% reporting successful births over the past five years [6]. This information signals a potential positive trend towards creating long-term sustainable captive populations of elephants in Thailand, at least in the North. Still, it will be difficult to estimate long-term population survival without data on longevity and death rates, which today are lacking.

The time required to register an elephant used to be at 8 years of age. However, this system required only recording the general appearance of each elephant on a registration card and having a microchip number. These loose controls allowed falsification and transfer of microchips between elephants, including those from the wild [8]. So, today, registration within 90 days of birth is mandatory, which may help to prevent the illegal capture of wild elephants for use in tourism.

Two welfare laws protect domestic animals, including captive elephants in tourism: the Criminal Code B.E.2499 (A.D. 1956) (section 381 and 382) and the Prevention of Cruelty and Animal Welfare Provision Act B.E.2557 (A.D. 2014). Welfare criteria are vaguely defined, however, and maximum fines too low to have a major impact. Thailand began to respond to criticisms about elephant welfare beginning in 2002 when the DLD created a set of standards and regulations for proper housing; food and water provisioning; health care and registration; mahout (i.e., keeper, handler) rights; facility environmental and waste management; and tourist services and safety. In 2009, additional standards were developed by the Ministry of Tourism and Sports in the Department of Tourism. Camps are required to be inspected and certified every two years. However, this is rarely enforced. Most camps do not comply because there is little incentive to participate and no penalty if they do not. Furthermore, certification does not impart any benefits like increased governmental support, income, additional veterinary services, or advertising promotion. As such, only six camps have been certified by the Department of Tourism and nine by the DLD, from approximately 250 camps in Thailand.

The Cruelty Prevention and Welfare of Animals Act was passed in 2014 to prevent animal cruelty and promote good welfare. It states that “no person shall perform any act which is deemed an act of cruelty to animal without justification” (Section 20), and that “an animal owner shall provide proper welfare to his or her animal in accordance with rules, procedures, and conditions prescribed by the Minister” (Section 22). The problem again is enforcement, which is exacerbated by the lack of defined welfare criteria. Thus, stronger and more enforceable regulations for the proper treatment of elephants in tourism are still needed and should be evidence-based. Even more recently, a sub-regulation of The Cruelty Prevention and Welfare of Animals Act on elephant welfare management has been completed and will be released for enforcement soon. Finally, welfare guidelines for Thai elephants in tourism have been created by the Faculty of Veterinary Medicine, Chiang Mai University (CMU) with input from governmental and non-governmental organizations, veterinarians, and researchers working with elephants, and will be published by July 2020.

## 3. Challenges Facing Elephant Tourism 

There are a number of obstacles that hinder efforts to ensure the welfare of captive elephants in Thailand: limited welfare knowledge of mahouts and camp/elephant owners; lack of enforceable welfare standards and laws; and decreasing mahoutship, all of which vary among camps and between regions. Welfare concepts such as the five freedoms [9] and five domains [10] models that drive management and care of elephants in Western zoos are not well understood or heeded by elephant managers in Thailand. A collective effort by all stakeholders is needed to establish stringent welfare standards and ensure that all camps comply through a certification process. But standards must be based on empirical evidence and not anecdote or emotion, the lack of which has empowered organizations to make unfounded claims of universal brutality towards elephants in tourism. What the industry needs is guidance from governing bodies that can inform on best practices for elephants, such as that provided for accredited Western (e.g., American Association of Zoos and Aquariums, British and Irish Association of Zoos and Aquariums, World Association of Zoos and Aquariums), and Eastern (e.g., South Each Asia Zoo Association, Zoological Park Organization) zoos. However, guidelines are only as effective as their enforcement, and good welfare is not guaranteed because guidelines exist. Therefore, an enforceable set of laws and standards that are stringently enforced, are desperately needed in Asia. 

Decreases in mahout experience and knowledge are major concerns for elephant camps today, and have been for decades [3,11,12,13]. As reported by Lair [3], a notable decline in skills has occurred within one short generation. In the past, the job of a mahout was often passed from father to son, and there was pride in the profession. Today, fewer young men are following in their father’s footsteps, and instead appear to be seeking higher paying jobs outside the field [14]. The job of a mahout is difficult. There are significant risks to safety, the job does not confer any social status or prestige, and the income is low [3]. When better paying work becomes available, mahouts often move from camp to camp, and this high turnover can lead to poorly developed skills and experience, and little bonding between mahout and elephant [11,13,15]. Furthermore, many mahouts belong to underprivileged ethnic groups (e.g., Karen, Tai Yai, and Palaung) within Thailand or neighboring countries, like Myanmar [6]. They often do not speak Thai, and so can have difficulty asking for better pay or health care. Similar concerns are apparent in Myanmar, where many mahouts today have insufficient experience, little familial connection to the profession, and change elephants often [12], all of which can compromise elephant welfare. It has been suggested that three years is needed for a mahout to understand their elephant’s behavior, and eight years to develop trust [16]. Many elephants respond only to their mahout [12,17], adding to problems associated with constant turnover. The welfare of an elephant is undoubtedly related to the welfare of its mahout given how important positive human–animal relationships are to animal welfare, including elephants [18]. Therefore, as discussed by Bansiddhi, et al. [6], there is a need to improve the quality of life for mahouts, including making sure they are paid a decent wage, receive appropriate training, especially in positive training techniques, and the profession becomes a source of pride.

In recent years, animal activist groups, international tour operators, and the public have begun questioning the welfare practices of elephants in tourism, with similar concerns levied against zoos and circuses [19,20,21,22]. Frequently asked questions and concerns include: Do elephants come from the wild? Are baby elephants separated from their mothers? How are baby elephants trained? How are working elephants trained for shows, rides, or painting? Are these activities cruel? Why are elephants controlled by hooks and chains? [23]. As a result, animal activists have created clear messages that use of an ankus (i.e., metal hook attached to a long handle, otherwise known as bullhook or guide [24]) and riding of elephants is always cruel, all elephants used for riding or shows have been brutally trained, and venues offering those activities should be boycotted [20,21]. In response, sites like TripAdvisor stopped selling tickets to venues where tourists interact directly with captive wildlife, including petting tigers, swimming with dolphins, and riding elephants [25]. National Geographic launched Wildlife Watch, an investigative reporting project focused on wildlife exploitation and the plights of several species, including captive elephants in Thailand [26]. Moreover, in December 2019, the UK’s largest travel association, the Association of British Tourism Agency (ABTA), updated its guidelines to advise members not to send customers to camps that allow direct contact between tourists and elephants, and states there should always be a barrier between elephants and people. In addition, elephant activities such as riding, bathing, and shows that include football or dancing performances are now considered unacceptable [27]. These campaigns are having a notable effect on Thailand’s tourism industry. Many camps now avoid riding, but rather than comply with the stated wishes of the international community, some camps simply hide the use of hooks and chains from the tourists [28]. The number of overweight and obese elephants is growing due to reduced exercise opportunities, and because in many camps the only way for tourists to interact with them is through feeding (e.g., bananas, sugar cane) [29,30]. While giving rides is one form of exercise, there are alternatives, such as allowing elephants to interact freely with their conspecifics, explore their environment, swim/bathe/mud wallow, and forage naturally, all of which can support good physical and psychological health [31,32,33]. 

One concern is how to ensure the safety of tourists, mahouts, and property if tools like the ankus are discouraged. What are the alternatives? Mahout injuries and fatalities are a concern if they cannot safely control elephants, which is one purpose of the ankus. Authors note the use of nails hidden in pockets and knives in mahout bags, which may be even more dangerous because the mahout needs be closer to correct an elephant than if an ankus, which extends the arm’s reach, was used [6]. However, it is not known how common or how big a problem this is. Finally, these restrictions can result in elephants being managed differently behind the scenes, with staff correcting misdeeds that occurred earlier or using reinforcing harsh training methods to ensure they behave during the day. All of this can create an unpredictable environment that results in poorer welfare than if equipment were allowed. However, it is important to acknowledge that elephants are often injured by misuse of physical equipment like the ankus [34], and so mahouts need to be instructed on its proper use, with consequences if rules are not followed. In addition, positive reinforcement training should be required to avoid or minimize the need to use an ankus. This type of training can be more effective in eliciting voluntary responses than negative reinforcement or punishment in a variety of situations, like during routine husbandry or veterinary procedures. Animals managed with positive reinforcement may also experience less anxiety or fear during training, thus contributing to a more cooperative relationship between staff and animal (see review, Kane et al. [35]). Protected contact, where handlers and elephants do not share the same space and interact with a barrier between them, emerged in response to concerns for keeper safety and animal welfare [36,37]. However, there can be limitations. Wilson, et al. [37] found longer lag times between verbal commands and elephant responses, and higher rates of refusals for elephants in protected versus free (i.e., elephants and people share the same space) contact. Protected contact confers more choice and control for elephants, but a risk is not being able to elicit immediate responses during health or safety emergencies if an elephant refuses to comply. Protected contact systems also require specialized facilities, stronger enclosures, and staff skilled in this type of training on an ongoing basis [38]. Thus, more studies are needed to determine how handling and training methods affect elephant welfare—both positively and negatively.

## 4. Studies Conducted on Welfare of Tourist Elephants

Welfare studies are beneficial for developing science-based guidelines and standards, which should not be based on human emotion or belief. To that end, for the past several years the authors have conducted a series of studies on tourist elephants throughout northern Thailand to determine what management practices affect health and welfare using an epidemiological approach (see review in this issue, Brown et al. [39]). The first study was a survey of elephant camps (*n* = 627 elephants at 33 camps) to quantify what activities elephants were engaged in, and their general management and care [6]. A subset of those elephants that could be blood sampled were further studied to assess metabolic markers (glucose, insulin, glucose to insulin ratio, fructosamine) and lipids (low and high density lipoproteins, total cholesterol, triglycerides) in relation to diets, physical activity, and levels of obesity [29,30,40]. Physical examinations quantified body condition, foot health, and skin wounds as physical welfare outcomes [34]. Fecal samples were collected to measure concentrations of glucocorticoid metabolites (FGM) as a physiological outcome, with presence or absence of abnormal behaviors (i.e., stereotypies) as a behavioral outcome [31]. The survey study found that elephant management and care varied considerably among camps, and may be trending away from activities like elephant shows and riding with saddles, to experiences like feeding, bathing, and walking [6]. As summarized in Figure 1, elephant camps can improve welfare by ensuring elephants have sufficient exercise and are fed limited amounts of high calories treats, especially in the high tourist season, to avoid problems with high body condition and unhealthy alterations in lipid profiles and metabolic function (i.e., glucose and insulin). Limited use of concrete floors and walkways can prevent foot and nail problems. Equipment for control (e.g., ankus) should be used properly to prevent injuries and unnecessary wounding. Providing more naturalistic housing conditions, opportunities to exercise, and controlling the number of tourists per day may reduce stress [6,29,30,31,34,40,41].

While some findings met our expectations, others did not. For example, elephants at camps that were observation only exhibited higher FGM concentrations than those where riding with a saddle and shows were allowed [29,31]. Only a limited number of elephants were examined in those studies, however, so this needs to be further explored. Glucocorticoids are commonly used as a stress marker and increase in response to a variety of intrinsic and extrinsic stimuli [42]. However, they do not unequivocally equate to welfare. In elephants, increases in glucocorticoid hormone or metabolite concentrations are observed in animals kept under poor husbandry conditions (see Bansiddhi, et al. [5]), but also follow normal physiological patterns (e.g., estrous cycle, musth) [43,44]. Activities like riding, walking, or shows can be enriching and may help prevent boredom, provide opportunities to socialize, and promote exercise to avoid obesity [40]. Elephants in observation and no riding camps generally spent little time socializing with other elephants, more time standing, and less time engaging in exercise compared to those in riding programs [6,31]. Walking distance, walking time, and working hour were all associated with lower FGM [31]. No riding camps can provide good welfare if managed properly (e.g., provide walking, enrichment and social opportunities). It is important then to evaluate all aspects of an elephant’s life collectively to try and tease out what factors have the biggest impact on physical and mental health.

Our data do support concerns that improper use of equipment to control elephants can negatively impact welfare in some elephants at some camps. For example, 27% of elephants controlled by an ankus had wounds in areas where the ankus is used, and higher wound scores were associated with higher FGM concentrations [31,34]. However, in the larger univariate or multivariable models, ankus use was not correlated to FGM concentrations [31], so these relationships are not straightforward and need to be explored further. Regardless, misuse of equipment must not be tolerated, and it is essential that mahouts receive proper training in positive control techniques, and that camps establish and/or enforce policies against use of unnecessary and punitive force. It is important to understand that engaging in activities with elephants is not always safe and can be life-threatening to mahouts, tourists, and elephants alike. The ankus is considered an important tool for mahouts managing elephants in free contact [24]. It is used to apply strong, clear pressure to particular control points that the elephant has been trained to react to [24]. One newer approach, based on techniques developed in horses, involves both ‘reward’ and ‘pressure-release’ [45]. Light pressure by a stick or ankus is applied and increased only if the elephant does not respond, and then is released when the elephant displays the desired behavior. In this way, the ankus can be an efficient tool for eliciting desired behaviors, and its use is not always punitive [45]. Even if barriers are placed between tourists and elephants, mahouts are still an important part of their care and must work safely in close contact. Simply eliminating the ankus may not be a panacea for ensuring good elephant welfare.

## 5. Five Domains Model—So How Well Do Camps Comply?

Based on authors’ experience and work in elephant camps for medical management over the past decade, welfare of tourist elephants in northern Thailand is improving, and many camps are actively working to establish better management protocols. We approach welfare assessments using the five domains model and find considerable variation in how well camps are complying. As reviewed in Mellor and Beausoleil [46], the first three domains focus on physiological and pathophysiological effects related to nutrition, environment and health, while a fourth considers external physical, biologic and social conditions that may affect the expression of normal behaviors or otherwise create significant challenges [47]. Then, the affects created by factors in these domains are placed into a fifth domain to evaluate mental health, and it is these measures combined that define an animal’s welfare state [47]. Here, we briefly highlight ways camps are or are not meeting elephants’ physical, physiological and psychological needs based on personal observations and research conducted by the authors [6,29,30,31,34,40].

### 5.1. Nutrition

Water is generally provided by mahouts during the day and available ad libitum in some, but not all camps. Few camps provide water at night, however [6]. Most camps have a policy to increase the frequency of water provisioning during hot weather. Elephants in the wild spend large percentages of time foraging [48], so roughage is fed regularly by mahouts and often supplemented with treats (bananas, sugar cane) during the day by tourists. In Thailand, although information on roughage fed was obtained (116 kg for adult males and 125 kg for adult females), amounts could have been underestimated because it was not weighed nor was intake determined. All elephant camps provided a variety of supplements, while over a third permitted free-foraging periods [6]. In a recent study, average body condition score (BCS) of elephants in northern Thailand generally was high, between four and five (out of five), meaning most were overweight; none had a BCS of 1 (too thin) [34]. Diets are somewhat limited, and elephants do not have access to the greater food choices of wild elephants [49,50]. Most camps feed two varieties of roughage (napier grass and corn stalks), with 2–3 varieties of supplements (e.g., banana, sugarcane, pumpkin, tamarind). About 39% of camps tether elephants in a nearby forest or grass field for 6–12 h a day to try and meet their need to free-forage [6]. In a survey by Chatkupt et al. [51], elephants in Chiang Mai had better body condition (i.e., were not too thin) compared to elephants in Bangkok and Phuket, which was attributed to better forage quality. Twenty years later, there are still concerns about some camps not providing adequate food and water to elephants. Few camps provide water at night, and many elephants do not have ad libitum access to food and water during the day [6]. Furthermore, tourist feeding of high calorie supplements is resulting in many elephants having overly high body condition that may be negatively impacting metabolic health [29,30,40]. Obesity also can have negative effects on physical health and reproduction [52], although correlations between BCS and musculoskeletal problems were not evident in the U.S. population [53].

### 5.2. Environment

There is generally good ventilation and fresh air at the elephant camps because of the more natural surroundings and open-air housing conditions. Most (88%) elephants are provided shelter (sheds, enclosures) to protect them from sunlight and rain [6]. No cases of hyperthermia have been observed by our team, although it does occur, albeit rarely and mostly in calves [54]. In summer, mahouts increase the frequency of bathing, and camps reduce the duration of or curtail elephant riding during the hottest part of the day. In winter, elephants may be warmed by blankets or bonfires, and nets may be put up to reduce cold wind exposure. None of the camps in our studies maintained elephants on concrete all of the time, and of those that used concrete flooring (about half), it was mainly during the day with dirt flooring provided in night rest areas [6]. A few camps provided sand stalls at night, which decreased the risk of skin wounds [34]. Replacing concrete with dirt or sand flooring is important to maintain good foot and joint health, and especially for old animals, softer substrates can encourage them to move, lie down and rest more comfortably [55]. Camps in northern Thailand are located outside of major cities, thus avoiding many types of associated pollution. However, there are no policies controlling tourist numbers, and in many camps, elephants may be exposed to excessive noise or unpredictable interactions, leading to compromised welfare that should be addressed.

### 5.3. Health

Common diseases or other health disorders include wounds, ectoparasites, nail cracks, constipation, colic, and eye problems, which generally are caused by improper management. Most wounds are caused by inappropriate use of the ankus [6,56]. Excessive walking on concrete and aging were related to foot problems, particularly nail cracks [34]. Gastro-intestinal problems have occurred by eating food contaminated by insecticides or fertilizers [56]. Metabolic problems associated with being overweight have been documented over the past five years from an imbalance between physical activity and dietary caloric intake [29]. Only 18% of camps have a veterinarian on site, although 36% have an elephant clinic, a specific area that has restraint facilities and basic medical supplies [6]. Many camps employ staff that have passed a veterinary assistant training course. All camps are visited at least twice a year by veterinarians from the National Elephant Institute (NEI), CMU, and/or the DLD National Institute of Elephant Research and Health Service, who conduct routine health exams and provide deworming services. Elephants also are screened annually for tuberculosis by the DLD. Elephant mobile clinics are provided by the NEI, CMU, and the Thai Elephant Alliance Association. There are two elephant hospitals in Lampang province, one private and one government-managed, that offer free veterinary care.

### 5.4. Behavior

At most camps, elephants are controlled primarily by mahouts during the day. Many elephants have opportunities to socialize with conspecifics when not involved with tourist activities, and we are encouraged that these opportunities are growing. During the night, elephants generally are isolated and kept on chains ranging from six to nine meters. While elephants are chained, some have the ability to interact with others nearby, but many do not and so can experience poor welfare. One behavioral indicator that welfare is suboptimal is the expression of stereotypic behaviors. Based on a yes/no question posed to mahouts by the lead author (a clinical elephant veterinarian), our camp survey [31] found about a quarter of elephants exhibited repetitive behaviors such as swaying, rocking or head bobbing. We suspect this may be an underestimation due to the lack of mahout knowledge of what stereotypic behavior is or what it means, and also to the limited amount of time each elephant was directly observed by the veterinarian. It also is possible that elephants may not stereotype as much when mahouts are present, so a follow up study is planned to utilize longer direct observation periods and an ethogram. Risk factors for developing stereotypies in captive elephants have been associated with long periods of chaining in particular [57,58,59,60,61], but also to being kept in small or unenriched spaces [32,62,63] and being socially isolated [32,61,64,65,66]. Some stereotypic behaviors are anticipatory and occur before routine provisioning of food and water, or before scheduled training or show performances [58,67,68]. Others may be displayed as a means to self-soothe [69]. In our study, elephants exhibiting stereotypic behaviors had lower FGM concentrations compared to those that did not [31]. It is important to note, however, that stereotypic behaviors may not reflect current conditions [69,70], but rather past environments that were suboptimal. Regardless of cause, repetitive behaviors indicate problems exist, so research is needed to identify ways to mitigate them and understand underlying driving conditions. 

A few camps do not chain elephants at all, but keep them in enclosures at night, although usually in isolation and with limited space (6 × 6 to 12 × 15 m), an area in need of improvement. In some camps (39%), elephants have opportunities to explore and forage, largely dependent on available fields or forest areas nearby. At issue is the increasing numbers of camps that are established each year in areas with limited natural habitat to contain them. Government regulations might help alleviate this pressure if permits to access national forests were enforced and numbers of camps limited. Elephant breeding is promoted in many camps in an effort to sustain the captive population [71], although a problem is the lack of either genetic or demographic management. Rearing young can be beneficial to female elephants, both physiologically [72] and behaviorally [73], and thus can be good for welfare. Providing proper nutrition and adequate facilities, and having staff experienced in breeding management is key to an ethical breeding program that works towards improved welfare for generations of captive elephants.

### 5.5. Mental State

Affects generated by factors in the first four domains are accumulated into the fifth domain to assess mental state [47]. Of all the domains, this is perhaps the most difficult to objectively determine. Elephants are intelligent, social animals, so ensuring that captive environments meet psychological needs is imperative, but not always possible in a tourist camp environment. Only some camps consider the mental state of elephants, including the basic need for socialization, or perhaps even understand that it is important. Based on our survey [6], newer camps are moving away from scheduled, primarily solitary activities to allowing more free time for elephants to interact and play, especially around bath time. More opportunities to socialize can result in better breeding success, with added welfare benefits associated with motherhood [74]. In the wild, elephants live in matriarchal societies, although Asian elephants show noticeably less social connectivity at the population level compared to Africans [75]. In a large-scale study of Myanmar timber elephants, there were clear reproduction and fitness benefits from strong matriliny relationships with sisters and grandmothers [76,77]. A number of factors shape the success of social groups, including choice and compatibility, size and composition, relatedness, social hierarchy stability, and individual personalities (reviewed by Williams et al. [78]). Unrelated tourist elephants rewilded into protected areas in Thailand formed bonded groups when calves were present [79]. Even if animals are unrelated, positive social interactions can provide a buffering effect against stressful challenges, and positively affect health and wellbeing [80,81,82].

Mahouts also play an important role in the life of elephants, both positive and negative. They can incite fear (e.g., punishment for misbehavior) or promote strong, caring bonds over time. From our survey, mahouts have been with their current elephant for three years on average (range, three months to 20 years, *n* = 61). In Myanmar, a limited study of mahouts (*n* = 10) found the average time with their elephant was 13.2 years, ranging from six months to 30 years [83]. Ultimately, the welfare of individual elephants is inextricably tied to the experience of its mahout, which unfortunately appears to be declining throughout Asia [12,83]. A study of zoo elephants found that positive keeper attitudes predicted lower mean serum cortisol in elephants [18]. In addition, the strength of keeper-elephant bonds predicted keeper job satisfaction. In the same way, positive mahout-elephant bonds undoubtedly have welfare benefits for both mahouts and elephants.

## 6. Welfare-Related Activities in Thailand

### 6.1. Asian Captive Elephant Working Group (ACEWG)

The ACEWG [84] was created in 2015 by a group of regional and international Asian elephant experts, including veterinarians, conservationists, and researchers. The group’s goal is to create awareness about problems facing elephants in tourism and provide recommendations for how to improve elephant management practices and health care throughout Asian range countries [85]. All of the authors are members of this group. It was formed to provide an objective voice to the intense debate about common concerns raised about elephant tourism practices. We aim to use objective and empirical research to set standards for captive Asian elephant management, promote good business practices with high welfare standards, and secure a sustainable quality of life for captive elephants throughout southeast Asia. Several documents are posted on the website to educate the public about captive elephant welfare, including 15 frequently asked questions on elephants in tourism, which is available in seven languages. The website provides a forum for posting research results and member activities as they pertain to elephant welfare. The group also facilitated the creation of a comprehensive set of camp standards focusing on elephant husbandry and care that are now being applied to a certification program for elephant camps located in Thailand and several other Asian countries (see below).

### 6.2. Thai Elephant Alliance Association

The Chiang Mai Elephant Alliance was formed in 2017 by a group of elephant camp owners in the region. The group meets regularly to discuss elephant issues faced by the membership, including how to address criticisms by animal activist groups, and ensure elephant welfare is a top priority. In August 2018, the name was changed to the Thai Elephant Alliance Association (TEAA), which today has members from 15 camps in northern Thailand, with hopes to expand to elephant camps throughout the country. The objectives are to: (1) promote member unity and work together on elephant-based tourism issues, (2) encourage positive perspectives about captive elephants and the way of life with elephants in Thailand, (3) be a support center for education and consulting about elephant-related topics, (4) help members provide good veterinary care, including supporting a mobile veterinary clinic, and (5) encourage and improve elephant conservation messages used by member facilities and others. The goal is to ensure all camps provide the best care for elephants based on objective welfare criteria and utilize the latest techniques in elephant management.

### 6.3. Development of Camp Standards

The lack of enforceable standards or laws specific to elephant welfare has hampered efforts to reform camps with substandard practices. In 2019, the Asian Captive Elephant Standards (ACES) was established to improve elephant welfare in South East Asia through certification of camps based on husbandry guidelines [86] created in consultation with several organizations: TravelLife, ACEWG, CMU, NEI and the Pacific Asian Tourism Association (PATA). The Tourist Authority of Thailand (TAT) and several tour operator companies now support this standard and will recommend tourists to ACES-certified camps. The certification process is voluntary, but it is hoped more camps will participate if it can increase tourist bookings, particularly from western countries. The camp standards will continue evolve as information becomes available on best practices, leading to further improvements in care and welfare.

### 6.4. Collaboration with Tourism Organizations 

The TAT’s mandate is to support the tourism industry in Thailand, including activities related to elephants, and so they see the value of an objective tourist camp certification system. They further recognize the urgency in helping tourists make informed decisions about what camps to visit, and where to find reliable information. A reduction in visitors means camps can face budget cuts, which would then reduce the living conditions of elephants and mahouts [87]. The net effect likely would mean poorer welfare through possible reductions in food quantity or quality, housing infrastructure or veterinary care. Our goal is to support good camps that have effective welfare programs and encourage those that do not to work towards incorporating more ethical practices. The criteria should be more than whether a camp offers elephant interactions or not.

The TAT is now collaborating with elephant researchers and veterinarians at CMU, NEI and the TEAA to promote elephant tourism through endorsement of camps that are certified and meet welfare standards. TAT also has agreed to promote education and conservation messages pertaining to elephants, and to help tourists be better informed. In 2019, TAT sponsored a well-rounded panel of elephant experts, and key stakeholders (e.g., ABTA, PATA) partook in an open discussion on the topic of “Elephant Wellbeing and the Thai Community” at the World Travel Market in London. The objective was to foster an understanding of the historical background of elephants in Thai culture, the varying stakeholder opinions regarding human-elephant relationships, and the best way forward for elephant welfare and tourism. Later, a number of international tour operators and media outlets were invited to visit several elephant camps with a variety of activities and management practices, to understand more about this controversial topic. The outcomes were fruitful, and several representatives expressed a change in their perception about elephant tourism in Thailand. More meetings are planned, which will hopefully continue to unite people from different sides of the elephant welfare discussion.

### 6.5. National Elephant Conservation Action Plan

In 2018, a 20-year National Elephant Conservation Action Plan (2018–2037) was developed, which includes four major strategies: (1) policy and regulations pertaining to elephants, (2) understanding socio-politics and culture related to elephants, (3) wild elephant conservation and habitat management, and (4) elephant welfare and health care. The goal of this plan is to improve health care of elephants across Thailand, which will entail establishing regional health care centers, and conducting training workshops in elephant health and welfare for mahouts, veterinary assistants, and camp owners.

### 6.6. Welfare Training Course and Workshops

Human resource development is key to enhancing welfare of elephants in Thailand. The Center of Elephant and Wildlife Research, Faculty of Veterinary Medicine, CMU has been conducting training courses since 2015 aimed at four audiences: mahouts, veterinary assistants, veterinarians; and biologists. At the mahout level, regional training is conducted in collaboration with the NEI and TEAA, and topics cover basic health care, as well as animal welfare concepts relevant to elephants. Generally, daily health checks are provided by the mahouts, the people closest to the elephants, and in the best position to detect problems early. Mahouts may not always recognize specific conditions however, so conducting health workshops is imperative, as is having a good relationship with the elephant. 

Training courses for veterinary assistants have been conducted in collaboration with the NEI since 2016. Chief mahouts are targeted for more extensive training in first aid and basic health care; e.g., wound cleaning, biological sample collection, drug administration, etc. under a veterinarian’s supervision. Welfare concepts also are taught, and as chief mahouts, this information is expected to be passed down to mahouts under their supervision. Veterinary assistants also play a key role in the early detection of health and application of emergency care, particularly in remote areas away from veterinary services.

International training courses and workshops for veterinarians and biologists have been conducted yearly to every other year since 2010 in collaboration with CMU and the NEI (Thailand), University of Peradeniya (Sri Lanka), Royal Veterinary College and Zoological Society of London (UK), and Smithsonian Conservation Biology Institute (USA), with the support of Faculty of Veterinary Medicine, Kasetsart University, British Council and Brian Nixon Fund for Protection of Elephants in Thailand. The curricula cover topics related to elephant biology, health care, breeding management, and welfare with lectures and hands-on practicals. To date, more than 100 people from around the world have participated in six courses. Most recently, to raise more awareness about elephant care and welfare, the NEI, Copenhagen Zoo, and Danish Animal Welfare Society conducted an international conference on Asian elephant welfare in 2018 at the NEI. Thailand, therefore, has engaged in both local and international educational efforts in health, management and welfare of captive elephants.

Finally, there is more interest now in using positive reinforcement training techniques as a result of several workshops sponsored by the Golden Triangle Asian Elephant Foundation (GTAEF), NEI, Zoological Park Organization, Elephant Kingdom-Surin, Africam Safari, and Human–Elephant Learning Programs (H–ELP). To date, four workshops have been conducted in Thailand, three in Myanmar, and one each in Laos and Cambodia.

## 7. Recommendations for the Future

We are beginning to understand how management affects the health and welfare of tourist elephants in northern Thailand through analyses of physical and physiological outcomes. These studies now need to be expanded to include other regions of Thailand and include more camps in each activity category. FGMs can inform on the well-being of animals, although it should be done in combination with other physiological, health, and behavioral measures. Difficult as it will be, we need to tease apart the influence of all factors in the captive environment that could affect welfare as they pertain to the five domains. From our elephant welfare studies, the important of proper diets, adequate exercise, natural and stimulating environments, and freedom of movement are key to good physical and psychological health and well-being [39]. Ultimately, it remains to be determined if ‘no hook’, ‘no riding’ policies are all that is needed for good welfare.

There is a need for further studies of behavioral outcomes. Stereotypies are among the most common welfare-related behaviors in captive elephants, and manifest in response to frustration or an inability to cope with challenges in the environment, or in anticipation of regularly scheduled activities, like work, training or feeding [88]. More systematic evaluations of factors related to the development or performance of stereotypic behaviors in tourist elephants are needed, similar to studies of elephants in western zoos [32,64], with direct observations conducted over long periods of time throughout the day. In particular, it is important to understand how boredom and chaining likely contribute to development of abnormal behaviors (e.g., Gruber, et al. [59]). Empirical investigations then can be conducted to determine how changes in management affect expression of repetitive behaviors so that mitigating strategies can be implemented. However, it is also important to understand that these behaviors may have developed in response to past environments that were suboptimal, and may not be reflect current conditions [69,70].

Current studies in Thailand are evaluating the usefulness of immunoglobulin A (IgA) measures as a possible indicator of wellbeing [89]. It is an immune protein associated with pathogen defense, but also has been shown to decrease during stress and increase in response to positive experiences [90]. IgA has been measured in elephants [89,91], although in contrast to some species [92,93], no correlations between patterns of salivary cortisol and IgA have been found [89,91]. Another approach that has recently been applied to wildlife is the use of allostatic load indexes, a method developed to better understand and describe the cumulative impacts of stress, particularly chronic stress, on an organism, and which can be used to advise on morbidity and mortality risks [53]. Elephants are long-lived, like humans—upon whom the allostatic load was based, so it could prove to be a powerful conservation tool for identifying factors in the captive environment that affect survival and fitness. More holistic approaches are needed to evaluate welfare in this complex species, involving combined assessments of health, behavior, and physiological function.

We need to better understand how separation of baby elephants from mothers for training affects welfare in the short- and long-term. The “Phajaan” ceremony performed in the north usually is conducted before training begins at about three years of age and believed to prevent bad spirits from harming the baby. The training takes about three months, during which time the calf learns to follow a mahout’s basic commands [94]. Harsh training methods are beginning to be replaced by more positive reinforcement techniques in part through international training workshops. In northern Thailand, there are some camps that train their own baby elephants, while others send them to the NEI, where more positive methods are now being applied. No studies have been conducted to quantify physiological or psychological impacts of training on a young calf or what the long-term consequences are of positive versus negative methods are, but they could be significant. In Myanmar, mortality risk of calves is highest during the first two years of life, followed by another rise between 4–5 years of age when calves are weaned and traditional training begins [95]. This situation is in contrast to natural behaviors in wild elephants, particularly with regards to calves, where female offspring remain with their natal herd for life and males emigrate at puberty [48]. While elephant behavior is usually discussed in workshops, more training by animal behaviorists is needed to emphasize the importance of understanding individual elephant behavior in the context of welfare. A new elephant behavioral assessment tool for routine use by keepers was recently published by Yon et al. [96] and could be useful for routine welfare monitoring of elephants in tourist camps.

One question raised by animal rights supporters is why can’t all captive elephants in Thailand be put back into the wild? In fact, more than 100 captive elephants have been released into several wildlife sanctuaries and national parks in Thailand as part of the HRH Queen Sirikit reintroduction project [97]. Although successful, with wild births recorded over the past 20 years, the project is now on hold because of concerns over increased human-elephant conflict, a growing problem in Thailand [98]. Some villagers have asked that the released elephants be removed from forest areas where they interfere with daily life. Moreover, increased human population growth, conversion of elephant habitat to agriculture, and resulting conflict means there currently is not enough wild space to release the ~3700 captive elephants in Thailand. A growing number of new approaches have been used to reduce the chances of subsequent human-wildlife conflict of rewilded animals, e.g., radio telemetry, sterilization and fertility controls, and modifying the behavior of problem animals long-term (see review, Nyhus [99], Snijders et al. [100]). Taking animal behavior into consideration is key for successful wildlife reintroductions, particularly that related to social behavior [101]. Understanding factors related to group cohesion and learning is important for selecting release animals that will be resilient, more likely to survive after release, and less likely to engage in conflict activities [101]. These methods should be applied to elephant reintroduction projects in Thailand.

Finally, captive elephants, although often referred to as ‘domestic’, have never been domesticated through selective breeding over generations for desired traits [5]. However, over the past decade, a number of camps have been purposely breeding elephants exhibiting non-aggressive and friendly behaviors (CT, PB, personal observations). It could be hundreds of years before significant changes in temperament are observed given the 20+ year generation interval. However, this may be one outcome of the tourist industry. Further research is needed to identify desirable behavioral traits and determine if selective breeding efforts to increase their expression in offspring is successful. It is critical that captive breeding efforts maximize genetic diversity and create stable population demographics, which would be aided by studbooks used by Western zoos, or logbooks like those for timber elephants in Myanmar (see Mar [102]). There are few examples of extensive documentation or record-keeping in Asia, which limits the ability to monitor demographic changes in health status or to assess factors affecting population morbidity, mortality and fecundity, all of which are important for developing effective management plans. Past studies have determined the genetic diversity of captive Asian elephants in Chiang Mai province is high [103,104], but continued attention is needed to ensure this variability is maintained, and to assess and monitor the genetic diversity of all elephants in the country [105].

## Figures and Tables

**Figure 1 animals-10-00919-f001:**
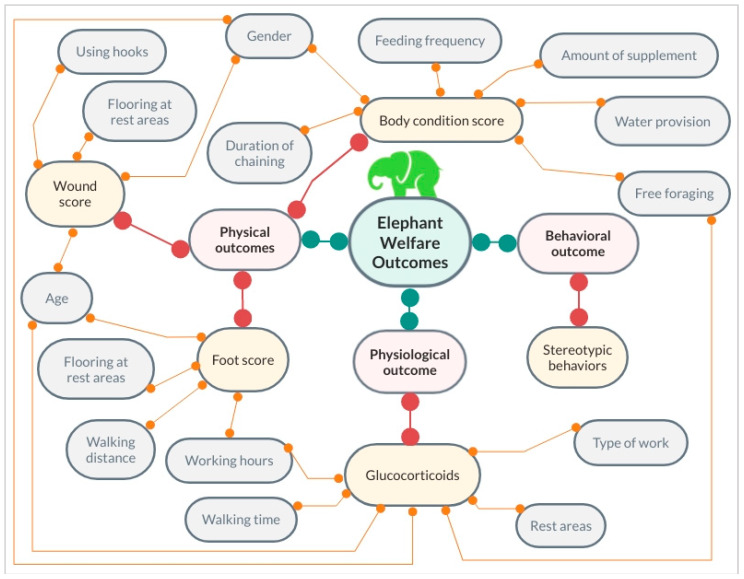
Associations between physical (foot score, wound score, body condition score), physiological (glucocorticoids), and behavioral (stereotypic behavior) outcomes and independent variables based on studies conducted on tourist elephants in Thailand [6,29,30,31,34,40,41]. Green lines indicate general outcome categories, red lines identify measured outcomes, and orange lines connect the independent variables to specific welfare outcomes.

**Table 1 animals-10-00919-t001:** Summary of significant events in the management and use of captive elephants in Thailand.

Year	Event
Late 13th century	Sukhothai kingdom: wild elephants caught and trained for war
Late 17th century	Ayutthaya kingdom: wild elephants caught and trained for war
Late 18th century	Rattanakosin era: wild elephants caught and trained for war
Late 19th century	Logging industry: wild and captive-born elephants employed for building roads and dragging logs to export sites
1921	Wild Elephant Protection Act: killing or capture of wild elephants prohibited
1939	Draught Animal Act: elephants considered private property
1969	Young Elephant Training Center established in Lampang
1970	Hill Tribe trekking tours established in northern Thailand
1972	Young Elephant Training Center established in Chiang Dao
1989	Logging prohibited by law
1990	Elephant camps established to provide work for former logging elephants
1992	Thai Elephant Conservation Center (TECC) established in Lampang (later change to National Elephant Institute (NEI))
2017	Captive population census updated using DNA fingerprinting
Today	Elephants mainly used in tourism, with some working in rubber plantations in the south

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
