# Peer review of "Welfare Assessment and Activities of Captive Elephants in Thailand"

_animals, 2020, doi:10.3390/ani10060919_

Round 1
Reviewer 1 Report
Thank you for your editing of the manuscript, it is much improved. However, there are still a few areas that would benefit from further editing/consideration, and particularly quite a number of places where a subjective opinion is given, and this needs to be rewritten to be more scientific and objective. Please see comments as detailed below:
Lines 60-62: the sentence doesn’t make sense as written, perhaps there are some words missing?
Line 82: “are well cared for…” suggest delete ‘well’ as this is a subjective assessment, and not appropriate.
Lines 97-98: suggest rephrasing so that the 6 and 9 are next to the two sets of dates listed (i.e. 6 campes in 2016 and 9 in 2017), to make this sentence easier to read.
Line 114: suggest ‘will’ instead of ‘would’
Line 133: need to change the word ‘appalling’, as this is subjective and emotional.
Similarly in line 134, change ‘massive’ to a more scientific and objective term (less colloquial).
Lines 135-136 “Nowadays, younger generations are more educated and most do not plan to follow in their father’s footsteps” again, this is written in a subjective way and needs to be phrased more objectively and scientifically
Lines 138-39: “The job of being a mahout can be daunting,” similarly rephrase to be more objective and scientific
lines 149-151: "These issues will not be resolved until mahouts are paid a decent wage, receive appropriate training, especially in positive training techniques, and the profession becomes a point of pride. " Again, this is a subjective assertion. This should be stated as an opinion, and a reference/citation provided.
Lines 179-182: this is an important point, but a citation/reference should be provided, so it is clear that this point of view is supported by the literature.
Lines 183-184: the authors state “"As these changes take hold, one problem is how to ensure the safety of tourists, mahouts, and property if tools are not allowed." This is not an accurate statement, as ‘tools’ in general have not been disallowed, but use of the ankus is frowned upon/discouraged. The authors should specify the ankus, and then can discuss why this may be a concern.
The authors cite a study in which latency between verbal command and elephants’ behavioural response was increased in protected vs. free contact, but have not discussed why this would be a problem – there is no clear welfare or safety concern discussed in relation to this issue.
In this paragraph discussing concerns about training and safety, there needs to be a more complete and informed discussion regarding the importance of good, and ongoing training for both mahouts and elephants (which hopefully would also decrease and ultimately eliminate the need for physical punishment as a training method).There should be a broader discussion of positive reinforcement training vs. physical punishment as a training approach and the evidence for/against these approaches.
Lines 225-226: It is unclear on what basis the authors make this assertion “Equipment for control (e.g., ankus) and restraint (i.e., chains) can be used, but only when employed properly”. The authors need to explain why they say that this equipment ‘can be used’. This was mentioned in a paragraph relating to elephant welfare measures, but it’s not clear how this conclusion was reached. This needs to be discussed, and full consideration given to alternative methods of control (i.e. better training of mahouts and elephants) to avoid or minimise the need to use this equipment.
In their assessment of the link between fGCM levels and whether elephants were used for rides vs. other activities, the authors concluded that camps offering ‘no riding’ may not guarantee a stress free environment, which is certainly a valid and important point. However, the authors then indicate that more research is needed, which is certainly true, but they should also consider/discuss other aspects of the elephants’ care at the non-riding camps which could also adversely affect their welfare. If, for example, when not being ridden or not involved in other activities, the elephants spend their time on restrictive chains, then giving rides may be the only time elephants have a break from that restriction, and thus may offer better welfare than those elephants unable to engage in that activity. This is just one possible explanation/suggestion, but the authors should consider what else could be impacting fGCM levels. And also the expression of stereotypies in elephants in these different management situations, as this is another important welfare indicator.
Lines 256-257: the authors again only refer to animal rights groups that have criticised ankus use; however, the criticism has been far wider than just animal rights groups. Please can the authors correct this.
In their discussion of the ankus (lines 256-262), the authors assert that the ankus is an essential management tool – please can the authors review the literature/evidence on this, and rather than providing a subjective opinion, provide evidence in support of the assertions that they make. If there is no evidence in support of this opinion, then this also needs to be made clear.
Lines 291-292: the authors refer to elephants that had ‘the best body condition’; please can the author provide more detail, as it is unclear what is meant by ‘the best’
lines 292-295:the authors assert " doing a reasonably good job of providing adequate nutrition and water"
This is a subjective assessment; how are they determining that it is 'reasonably good'? Since few camps provide water overnight, and many camps do not provide ad lib access to water at any time, this is arguably not 'reasonably good' and should be discussed further, as this is an essential aspect of good husbandry for good health. Moreover, obese elephants are by no means an indication of adequate nutrition, any more than obese humans that regularly consume a low quality 'junk food' diet have adequate nutrition, and this should be considered/discussed. There should be more discussion on the importance of provision of browse, for the opportunity to express increased foraging behaviour, better GI health, and decreased body weight
In terms of substrate/flooring, the fact that half the camps keep their elephants on concrete flooring during the day (which may be for 12 hours or more), is of substantial concern for foot and joint health, and this should be discussed. Furthermore, the fact that dirt floor is the only thing available at night is also a concern, particularly as the elephants may be very restricted in their ability to move/lie down. Moreover, for those elephants with foot or joint problems, there is no consideration/provision of sand and sand piles so that they can lie down more comfortably – this is particularly important for elephants with bad joints, as they will likely have difficulty moving into full recumbency.
There should be a discussion of the common health problems, and their potential connection to management/nutrition/husbandry issues (e.g. obesity and constipation related to poor diet [insufficient roughage] and insufficient opportunities for exercise)
Lines 322-324: the authors state: “Of all the southeast Asian elephant range countries, Thailand has perhaps the best veterinary capacity and network to care for captive elephants.”. This again appears to be a highly subjective assessment, and this sentence should be deleted.
The authors comment that rearing young is good for the welfare of female elephants, and this may be true, but the authors should also indicate that improvements in nutrition, management and facilities are important (and indeed owners are morally and ethically obligated to make these), in order to continue to work towards improved welfare for this next generation of captive elephants.
Line 382: “is comprised of” please correct this grammar
Line 423, the authors refer to elephant-related tourism being ‘under fire’ – this is not a scientifically appropriate phrase, suggest replacing with ‘under criticism’ or something similar, which is less emotive.
Line 425, the authors refer to: "how to separate fact from fiction, truth from hyperbole. ". please can the authors rephrase this using more scientific language.
Lines 428-430: the authors state “Our goal is to … not punish camps merely because they offer elephant interactions, some of which can have positive effects. “ again, this is phrased in a subjective way, and should be phrased in a more impartial and objective way.
Line 440, refer to ‘media from around the world’, please correct ‘media’
lines 440-441: "...visit several elephant camps known for good management and welfare, " Please rephrase this, as again this is a subjective opinion as currently phrased.
Lines 479-480: “Thailand, therefore, is at the forefront of local and international educational efforts in health, management and welfare of captive elephants. “ Please rephrase this, as this is again an opinion
in lines 490-492, the authors state: "Especially needed is a thorough evaluation of management practices at observation-only camps, given the surprising finding that FGM concentrations were higher than at those with more intensive tourist activities". However, while this is certainly a reasonable concern, FGMs are by no means the only indicators of welfare (and arguably have many substantial limitations), so many other considerations of welfare are also essential, which also need to be discussed by the authors, and include:
- use of chains
- inappropriate substrates (need to provide sand and sand piles, remove concrete entirely, provide softer substrates and not hard packed dirt, especially overnight)
- better diet
- mental and physical stimulation; access to socially compatible elephants (ideally multi-generational family groups), avoid social isolation of bulls, access to browse (ideally foraging opportunities),
- ad libitum access to water day and night
- access to water for swimming
The discussion of release of elephants into the wild, and the problem of human-elephant conflict should also consider the substantial body of literature regarding the methods that should be used in captive breeding/care for animals destined to be released, to reduce the chances of subsequent human-wildlife conflict. It is by no means inevitable that animals released into the wild will engage in human-elephant conflict.
The authors refer to efforts on selective breeding by camps in favour of non-aggressive behaviours. Rather than assuming that this effort has been successful (ie that the camps have successfully identified these traits, and that their breeding efforts have succeeded), the authors should propose further research both to create methods/tools to better identify desirable behavioural traits, and also to determine if selective breeding efforts to increase their expression in offspring is successful.
Moreover, the authors should advocate the use of genetic management of the population to assess and monitor genetic diversity in the population.
Reviewer 2 Report
This paper is much needed and does a good job of staying empirical for the most part. However, although I appreciate the authors including the reference I suggested during my previous review (Wilson et al., 2015), it seems as if what was included was cherry-picked from the abstract to support the authors' rational for using the hook (e.g. what was reported was that PR seemed less effective than FR), but the authors failed to mention that the results they reported were likely due to the animals' exercising more control and choice-based interactions, which is a positive welfare sign. I am disappointed in this, given that this paper calls for empirical discussion.
I also would have liked to see a little more emphasis on examining empirically the hook vs. no hook option and protected contact, which the authors argue does not allow for a close relationship with the mahout. Based on experience and personal observations, this does not seem to be the case. However, there is a need for both sides to empirically examine this in more detail.
Round 2
Reviewer 1 Report
Thank you to the authors for their work in trying to address the issues previously identified; overall, the manuscript is substantially improved. However, there are still quite a few areas where further work/clarification/correction is needed, throughout the manuscript.
Specifically:
Abstract:
Line 21: use of term ‘epicenter’ is somewhat exaggerated, and unsubstantiated. Need to rephrase
Line 21-22: refers to visiting the elephant camp as ‘one of the most popular attractions’. Again this is not supported with reference to externally verifiable sources. Need to rephrase
Line 40: say that elephant camps ‘mushroomed’ – suggest a more scientific and objective term
Line 42: again refer to Thailand as being the ‘epicenter’ of elephant tourism. Again, this is not substantiated with reference to externally verifiable sources. Please rephrase.
Lines 72-75: refer to percentages of camps reporting successful births, and state that this signals a positive trend towards creating a long term sustainable captive population. While this would potentially support this, need to also consider longevity and death rate in the captive population.
Lines 81-82: mention confiscation of unregistered elephants which are then kept indefinitely at a captive institute, rather than being released into the wild. This has potentially profound welfare implications for those elephants which should be discussed/considered.
Lines 200-202: the authors suggest that with protected contact, there is a risk in not being able to elicit an immediate response during health and safety emergencies. However, there should be a discussion as to what evidence there is that this has ever been an issue/problem (that health and safety emergencies have arisen and not having an ankus was a problem).
Lines 268-271: the authors assert that while some elephants with ‘more gentle temperaments’ can be handled without an ankus, others cannot. The authors need to cite evidence of this statement, as right now it seems to simply be their opinion.
Line 293: authors mention that >1/3 of camps permit free foraging periods. Please can they indicate where the elephants are allowed to forage, what is the type of forage (they are not all equivalent), and for what duration are the elephants allowed to forage (even if they can provide information on the range of time, and the average time they are allowed to do this)?
Line 305: the authors refer to high body condition having an adverse impact on metabolic health. However, obesity also has an adverse impact on joint health (and since arthritis is a substantial health issue in captive elephants this is very relevant, particularly as many are kept on very hard surfaces for a substantial portion of their day). Also obesity can impair reproductive success (libido, fertility, ability to copulate, etc), and this should also be discussed as it is also a significant concern/problem for captive elephants.
Line 311: the authors indicate that hyperthermia occurs ‘in other regions’; please can they specify or give examples of the locations?
Line 328: the authors refer to GI problems in relation to eating contaminated foods. But this can also be the result of eating a diet with insufficient roughage, which is an important consideration for elephants, as hind gut fermenters, who have evolved to have a diet high in roughage. This should be mentioned/discussed.
Line 329: again, the authors refer to obesity and metabolic problems. They should also discuss the adverse impacts of obesity on musculoskeletal health and reproductive activity.
Line 332: the authors mention that 36% of camps have an elephant clinic (as opposed to 18% having a veterinarian). It is unclear what is meant by an ‘elephant clinic’; please can the authors explain/describe this further
Lines 343-344: the authors indicate that the majority of elephants are kept isolated and on chains overnight. There are substantial welfare concerns associated with this, ranging from restricted ability to move, to forage, to freely interact with other elephants (beyond visual or auditory interaction at a distance), to comfortably lie down, etc. This should be clearly identified and discussed.
Lines 358-359: the authors mention an elephant camp survey – it would be helpful to indicate who completed this survey, and also what training they had in being able to recognise and identify stereotypies. If they did not have any training in this methodology, this should be explicitly acknowledged and discussed as a limitation in the data from the survey.
Lines 364-365: the authors suggest that more research is needed to identify ways to mitigate expression of stereotypies. While this is true, more research is also needed to understand what conditions are driving elephants to develop these behaviours; it would be helpful to articulate this as well.
Lines 374-375: the authors mention that increased opportunities to socialise would likely increase breeding success. While this is true, the opportunity to socialise has much broader and very important welfare implications, given that we know, from studies of wild elephant behaviour, the importance of social interactions to elephants. This should be mentioned/discussed as well.
Line 386: the authors mention that elephant welfare is tied to the ‘quality’ of the mahout. It is unclear what is meant by this – are they referring to the learning/training of the mahout? Need to rephrase for clarity.
Line 394: the authors suggest that work, including riding, may alleviate boredom. While that may be the case, the authors should also discuss that increased opportunities for elephants to interact socially with one another in open areas, and to swim/wallow would also increase physical activity/exercise and help alleviate possible boredom. Furthermore, work in the form of riding may be more stressful to the elephants, particularly if there are lots of tourists (in high season), or if they are physically abused by their mahouts, or don’t have a positive working relationship with their mahouts. All of this should also be discussed/considered.
Line 477: the authors indicate that mahouts are ‘the ones who know best when problems occur’ – this should be rephrased. They are perhaps in the best position to detect problems early, as they work regularly with their elephants. But they may not know how to recognise problems, so they must have training in order to do so, and need a good working relationship with their elephant. This should be rephrased, and also these other considerations discussed.
Lines 487-489: use the term ‘bi-yearly’ should be biennially. Also, mention ‘since 2010’ twice in that sentence.
Lines 508-509: the authors indicate that FGM concentrations were higher in elephants involved in more interactive tourist activities. Please can the authors provide a bit more detail to give this context? What were the differences in FGM levels (ranges and mean/median values?). And how many camps/types of activities were assessed?
Throughout the paper, the authors mention that they have used stereotypies as their measure of behaviour. However, as the authors briefly discuss at the very end of the manuscript, there are a whole range of other behaviours that should also be assessed when considering welfare. It’s good that this was mentioned/acknowledged, but this needs to be done at the beginning of the paper (for example in Fig 1 and the discussions of how welfare outcomes should be assessed), and throughout, as this is a significant deficiency in the current data.
Lines 533-534: the authors mention allostatic load indices, but describe them as markers to assess morbidity and mortality risks – this is not really accurate. This model was developed to better understand and describe the cumulative impacts of stress, particularly chronic stress, on an organism. This needs further explication by the authors.
Lines 539-555: the authors discuss the training methods of calves in Thailand, in the context of considering the welfare impacts of types of training techniques used with the calves. However, the other very significant feature of this way of training calves is the early separation of the calves from their mothers. This is in stark contrast to natural behaviours in wild elephants, as females often remain in their natal group their entire lives, and bulls only leave around 10-12 years of age when they reach puberty. This has profound welfare implications for these calves that are separated from their mothers at an early age, and should also be acknowledged and discussed.
Line 575: the authors suggest that it will be decades before changes are seen in elephants as a result of the selective breeding that has taken place in elephant camps in Thailand over the past 10 years. This is not a correct interpretation of the process of domestication, as this involves selective breeding over many generations. Given the lifespan of elephants, this impact would likely take hundreds of years to be achieved, so could only be realistically assessed after that time period. This needs to be corrected in the manuscript.
Overall, there were quite a number of places throughout the paper where the authors self-cite, or use their personal opinions as evidence in support of an assertion. It would be helpful if they can avoid this, and work to find published evidence in support of many of their assertions.
Author Response
Comments and Suggestions for Authors
Thank you to the authors for their work in trying to address the issues previously identified; overall, the
manuscript is substantially improved. However, there are still quite a few areas where further
work/clarification/correction is needed, throughout the manuscript.
Response to the comment: Thank you so much for your valuable comments. We have carefully edited the
manuscript. We hope you agree with the changes.
Specifically:
Abstract:
Line 21: use of term ‘epicenter’ is somewhat exaggerated, and unsubstantiated. Need to rephrase.
Response to the comment: We explained in our first revision that the term ‘epicenter’ is appropriate. Some
definitions are “the central point of something, typically a difficult or unpleasant situation”, “a thing or place
that is of greatest importance to an activity or interest”, and “the very center of something”, all of which
describes the tourist industry in Thailand. In addition, there are numerous references to Thailand being an
epicenter of elephant tourism, including the National Geographic
(https://www.nationalgeographic.com/travel/lists/world-elephant-day-where-to-go-what-to-know/) and Wall
Street Journal (https://www.wsj.com/articles/should-tourists-ride-elephants-in-thailand-1533914087), so we
feel justified using this term. We removed the word from the title, but believe it is still appropriate to use it
in the abstract.
Line 21-22: refers to visiting the elephant camp as ‘one of the most popular attractions’. Again this is not
supported with reference to externally verifiable sources. Need to rephrase
Response to the comment: We changed the sentence to “visiting an elephant camp is a popular activity
according to the Tourist Authority of Thailand.”
Line 40: say that elephant camps ‘mushroomed’ – suggest a more scientific and objective term
Response to the comment: It now reads “Beginning in the north, elephant camps have increased in number
throughout the country [Lair, 1997; Ringis, 1996].”.
Line 42: again refer to Thailand as being the ‘epicenter’ of elephant tourism. Again, this is not substantiated
with reference to externally verifiable sources. Please rephrase.
Response to the comment: See above where we address this comment.
Lines 72-75: refer to percentages of camps reporting successful births, and state that this signals a positive
trend towards creating a long term sustainable captive population. While this would potentially support this,
need to also consider longevity and death rate in the captive population.
Response to the comment: This now reads “One aim was to prevent the illegal capture of wild elephants
for use in tourism, which has put pressure on elephant venues to maintain elephant numbers through captive
breeding. In a survey of 33 camps in northern Thailand, elephants breed well in some, but not all camps,
with 55% reporting successful births over the past 5 years [Bansiddhi et al., 2018]. This information signals
a potential positive trend towards creating long-term sustainable captive populations of elephants in
Thailand, at least in the north. Still, it will be difficult to estimate long-term population survival without data
on longevity and death rates, which today are lacking.”.
Lines 81-82: mention confiscation of unregistered elephants which are then kept indefinitely at a captive
institute, rather than being released into the wild. This has potentially profound welfare implications for
those elephants which should be discussed/considered.
Response to the comment: This sentence has been deleted “Unregistered elephants not in the microchip or
DNA databases are confiscated, registered and kept at the rehabilitation center at the National Elephant
Institute (NEI) in Lampang indefinitely. These elephants are cared for by a team of elephant veterinarians.”.
Lines 200-202: the authors suggest that with protected contact, there is a risk in not being able to elicit an
immediate response during health and safety emergencies. However, there should be a discussion as to what
evidence there is that this has ever been an issue/problem (that health and safety emergencies have arisen
and not having an ankus was a problem).
Response to the comment: We offered a published reference for this statement (Wilson et al), but do not
feel it is appropriate to provide additional anecdotal testimonials. So no change was made.
Lines 268-271: the authors assert that while some elephants with ‘more gentle temperaments’ can be
handled without an ankus, others cannot. The authors need to cite evidence of this statement, as right now it
seems to simply be their opinion.
Response to the comment: This sentence has been deleted “While some elephants with more gentle
temperaments likely can be handled without an ankus, others cannot, and mahouts and camp owners must
know the difference and manage accordingly.”.
Line 293: authors mention that >1/3 of camps permit free foraging periods. Please can they indicate where
the elephants are allowed to forage, what is the type of forage (they are not all equivalent), and for what
duration are the elephants allowed to forage (even if they can provide information on the range of time, and
the average time they are allowed to do this)?
Response to the comment: The sentence now reads “About 39% of camps tether elephants in a nearby
forest or grass field for 6 to 12 hours a day to try and meet their need to free-forage [Bansiddhi et al., 2018].”
Actual nutritional analysis of these foraging areas have not been conducted so speculating on their quality
would not be appropriate.
Line 305: the authors refer to high body condition having an adverse impact on metabolic health. However,
obesity also has an adverse impact on joint health (and since arthritis is a substantial health issue in captive
elephants this is very relevant, particularly as many are kept on very hard surfaces for a substantial portion
of their day). Also obesity can impair reproductive success (libido, fertility, ability to copulate, etc), and this
should also be discussed as it is also a significant concern/problem for captive elephants.
Response to the comment: The section now reads “Furthermore, tourist feeding of high calorie
supplements is resulting in many elephants having overly high body condition that may be negatively
impacting metabolic health [Norkaew et al., 2018; Norkaew et al., 2019a; Norkaew et al., 2019b]. Obesity
also can have negative effects on physical health and reproduction [Brown, 2019], although correlations
between BCS and musculoskeletal problems were not evident in the U.S. population [Edwards et al.,
2019].”.
Reference:
Brown, J.L. Update on comparative biology of elephants: Factors affecting reproduction, health and welfare.
In Reproductive Sciences in Animal Conservation - Progress and Prospects. Advances in Experimental
Medicine and Biology, 2nd ed.; Holt, W.V., Brown, J.L., Comizzoli, P., Eds.; Springer Science and
Business Media: New York, NY, 2019, pp. 243-274.
Line 311: the authors indicate that hyperthermia occurs ‘in other regions’; please can they specify or give
examples of the locations?
Response to the comment: Fowler did not indicate what regions, so the sentence now reads “No cases of
hyperthermia have been observed by our team, although it does occur, albeit rarely and mostly in calves
[Fowler, 2008].”.
Line 328: the authors refer to GI problems in relation to eating contaminated foods. But this can also be the
result of eating a diet with insufficient roughage, which is an important consideration for elephants, as hind
gut fermenters, who have evolved to have a diet high in roughage. This should be mentioned/discussed.
Response to the comment: We are unaware of any published papers on GI problems in elephants being
related to insufficient roughage. If the reviewer can share that with us, we will incorporate it.
Line 329: again, the authors refer to obesity and metabolic problems. They should also discuss the adverse
impacts of obesity on musculoskeletal health and reproductive activity.
Response to the comment: See above where we address this comment.
Line 332: the authors mention that 36% of camps have an elephant clinic (as opposed to 18% having a
veterinarian). It is unclear what is meant by an ‘elephant clinic’; please can the authors explain/describe this
further
Response to the comment: This now reads “Only 18% of camps have a veterinarian on site, although 36%
have an elephant clinic, a specific area that has restraint facilities and basic medical supplies [Bansiddhi et
al., 2018].”.
Lines 343-344: the authors indicate that the majority of elephants are kept isolated and on chains
overnight. There are substantial welfare concerns associated with this, ranging from restricted ability to
move, to forage, to freely interact with other elephants (beyond visual or auditory interaction at a distance),
to comfortably lie down, etc. This should be clearly identified and discussed.
Response to the comment: This section has been rewritten “At most camps, elephants are controlled
primarily by mahouts during the day. Many elephants have opportunities to socialize with conspecifics when
not involved with tourist activities, and we are encouraged that these opportunities are growing. During the
night, elephants generally are isolated and kept on chains ranging from 6 to 9 meters. While elephants are
chained, some have the ability to interact with others nearby, but many do not and so can experience poor
welfare. One behavioral indicator that welfare is suboptimal is the expression of stereotypic behaviors.
Based on a Yes/No question posed to mahouts by the lead author (a clinical elephant veterinarian), our camp
survey [Bansiddhi et al., 2019] found about a quarter of elephants exhibited repetitive behaviors such as
swaying, rocking or head bobbing. We suspect this may be an underestimation due to lack of mahout
knowledge of what stereotypic behavior is or what it means, and also to the limited amount of time each
elephant was directly observed by the veterinarian. It also is possible that elephants may not stereotype as
much when mahouts are present, so a follow up study is planned to utilize longer direct observation periods
and an ethogram. Risk factors for developing stereotypies in captive elephants have been associated with
long periods of chaining in particular [de Mel et al., 2013; Friend & Parker, 1999; Gruber et al., 2000;
Schmid, 1995; Varadharajan et al., 2016], but also to being kept in small or unenriched spaces [Elzanowski
& Sergiel, 2006; Greco et al., 2017; Harris et al., 2008], and being socially isolated [Greco et al., 2017,
2016; Kurt & Garai, 2001, 2007; Varadharajan et al., 2016]. Some stereotypic behaviors are anticipatory and
occur before routine provisioning of food and water, or before scheduled training or show performances
[Friend, 1999; Rees, 2004, 2009]. Others may be displayed as a means to self-soothe [Mason & Latham,
2004]. In our study, elephants exhibiting stereotypic behaviors had lower FGM concentrations compared to
those that did not [Bansiddhi et al., 2019]. It is important to note, however, that stereotypic behaviors may
not reflect current conditions [Mason & Veasey, 2010; Mason & Latham, 2004], but rather past
environments that were suboptimal. Regardless of cause, repetitive behaviors indicate problems exist, so
research is needed to identify ways to mitigate them and understand underlying driving conditions.
A few camps do not chain elephants at all, but keep them in enclosures at night, although usually in
isolation and with limited space (6 by 6 to 12 by 15 m), an area in need of improvement. In some camps
(39%), elephants have opportunities to explore and forage, largely dependent on available fields or forest
areas nearby. At issue is the increasing numbers of camps that are established each year in areas with limited
natural habitat to contain them. Government regulations might help alleviate this pressure if permits to
access national forests were enforced and numbers of camps limited. Elephant breeding is promoted in many
camps in an effort to sustain the captive population [Thitaram, 2012], although a problem is the lack of
either genetic or demographic management. Rearing young can be beneficial to female elephants, both
physiologically [Hermes et al., 2004] and behaviorally [Prado-Oviedo et al., 2016], and thus can be good for
welfare. Providing proper nutrition and adequate facilities, and having staff experienced in breeding
management is key to an ethical management program that works towards improved welfare for generations
of captive elephants.”.
References:
de Mel, R.K.; Weerakoon, D.K.; Ratnasooriya, W.D. A comparison of stereotypic behaviour in Asian
elephants at three different institutions in Sri Lanka. Gajah 2013, 38, 25-29.
Friend, T.H.; Parker, M.L. The effect of penning versus picketing on stereotypic behavior of circus
elephants. J. Appl. Anim. Behav. Sci. 1999, 64, 213-225.
Gruber, T.M.; Friend, T.H.; Gardner, J.M.; Packard, J.M.; Beaver, B.; Bushong, D. Variation in stereotypic
behavior related to restraint in circus elephants. Zoo Biol. 2000, 19, 209-221.
Schmid, J. Keeping circus elephants temporarily in paddocks - the effects on their behaviour. Anim. Welfare
1995, 4, 87-101.
Varadharajan, V.; Krishnamoorthy, T.; Nagarajan, B. Prevalence of stereotypies and its possible causes
among captive Asian elephants (Elephas maximus) in Tamil Nadu, India. J. Appl. Anim. Behav. Sci.
2016, 174, 137-146.
Elzanowski, A.; Sergiel, A. Stereotypic behavior of a female Asiatic elephant (Elephas maximus) in a zoo. J.
Appl. Anim. Welfare Sci. 2006, 9, 223-232.
Greco, B.J.; Meehan, C.L.; Heinsius, J.L.; Mench, J.A. Why pace? The influence of social, housing,
management, life history, and demographic characteristics on locomotor stereotypy in zoo elephants. J.
Appl. Anim. Behav. Sci. 2017, 194, 104-111.
Harris, M.; Sherwin, C.; Harris, S. The Welfare, Housing and Husbandry of Elephants in UK Zoos: Final
Report; University of Bristol: Bristol, UK, 2008.
Greco, B.J.; Meehan, C.L.; Hogan, J.N.; Leighty, K.A.; Mellen, J.; Mason, G.J.; Mench, J.A. The days and
nights of zoo elephants: Using epidemiology to better understand stereotypic behavior of African
elephants (Loxodonta africana) and Asian elephants (Elephas maximus) in North American zoos. PLOS
ONE 2016, 11, e0144276.
Kurt, F.; Garai, M. Stereotypies in Captive Asian Elephants—A Symopsium of Social Isolation, Abstracts
for the International Elephant and Rhino Research Symposium, Vienna, Austria, 2001, pp 57-63.
Kurt, F.; Garai, M.E., The Asian Elephant in Captivity: A Field Study. Cambridge University Press India
Pvt. Ltd.: New Delhi, India, 2007.
Rees, P.A. Low environmental temperature causes an increase in stereotypic behaviour in captive Asian
elephants (Elephas maximus). J. Therm. Biol. 2004, 29, 37-43.
Rees, P.A. Activity budgets and the relationship between feeding and stereotypic behaviors in Asian
elephants (Elephas maximus) in a zoo. Zoo Biol. 2009, 28, 79-97.
Mason, G.J.; Latham, N.R. Can't stop, won't stop: Is stereotypy a reliable animal welfare indicator? Anim.
Welfare 2004, 13, 57-69.
Mason, G.J.; Veasey, J.S. How should the psychological well-being of zoo elephants be objectively
investigated? Zoo Biol. 2010, 29, 237-255.
Lines 358-359: the authors mention an elephant camp survey – it would be helpful to indicate who
completed this survey, and also what training they had in being able to recognise and identify
stereotypies. If they did not have any training in this methodology, this should be explicitly acknowledged
and discussed as a limitation in the data from the survey.
Response to the comment: This section now reads “Based on a Yes/No question posed to mahouts by the
lead author (a clinical elephant veterinarian), our camp survey [Bansiddhi et al., 2019] found about a quarter
of elephants exhibited repetitive behaviors such as swaying, rocking or head bobbing. We suspect this may
be an underestimation due to lack of mahout knowledge of what stereotypic behavior is or what it means,
and also to the limited amount of time each elephant was directly observed by the veterinarian. It also is
possible that elephants may not stereotype as much when mahouts are present, so a follow up study is
planned to utilize longer direct observation periods and an ethogram.”.
Lines 364-365: the authors suggest that more research is needed to identify ways to mitigate expression of
stereotypies. While this is true, more research is also needed to understand what conditions are driving
elephants to develop these behaviours; it would be helpful to articulate this as well.
Response to the comment: This now reads “Regardless of cause, repetitive behaviors indicate problems
exist, so research is needed to identify ways to mitigate them and understand underlying driving
conditions.”.
Lines 374-375: the authors mention that increased opportunities to socialise would likely increase breeding
success. While this is true, the opportunity to socialise has much broader and very important welfare
implications, given that we know, from studies of wild elephant behaviour, the importance of social
interactions to elephants. This should be mentioned/discussed as well.
Response to the comment: We have added “More opportunities to socialize can result in better breeding
success, with added welfare benefits associated with motherhood [Williams et al., 2019]. In the wild,
elephants live in matriarchal societies, although Asian elephants show noticeably less social connectivity at
the population level compared to Africans [de Silva & Wittemyer, 2012]. In a large-scale study of Myanmar
timber elephants, there were clear reproduction and fitness benefits from strong matriliny relationships with
sisters and grandmothers [Lahdenperä et al.; 2016; Lynch et al., 2019].”.
References:
de Silva, S.; Wittemyer, G. A comparison of social organization in Asian elephants and African savannah
elephants. Int. J. Primatol. 2012, 33, 1125-1141.
Lynch, E.C.; Lummaa, V.; Htut, W.; Lahdenperä, M. Evolutionary significance of maternal kinship in a
long-lived mammal. Phil. Trans. R. Soc. 2019, 374, 20180067.
Lahdenperä, M.; Mar, K.; Lummaa, V. Nearby grandmother enhances calf survival and reproduction in
Asian elephants. Sci. Rep. 2016, 6, 27213.
Line 386: the authors mention that elephant welfare is tied to the ‘quality’ of the mahout. It is unclear what
is meant by this – are they referring to the learning/training of the mahout? Need to rephrase for clarity.
Response to the comment: This now reads “Ultimately, the welfare of individual elephants is inextricably
tied to the experience of its mahout, which unfortunately appears to be declining throughout Asia [Crawley
et al., 2019; Mumby, 2019].”
Line 394: the authors suggest that work, including riding, may alleviate boredom. While that may be the
case, the authors should also discuss that increased opportunities for elephants to interact socially with one
another in open areas, and to swim/wallow would also increase physical activity/exercise and help alleviate
possible boredom. Furthermore, work in the form of riding may be more stressful to the elephants,
particularly if there are lots of tourists (in high season), or if they are physically abused by their mahouts, or
don’t have a positive working relationship with their mahouts. All of this should also be
discussed/considered.
Response to the comment: We have deleted the paragraph - Other concerns related to mental health include
boredom in camps with limited activities or social opportunities. Suspected increases in stress during the
high tourist season, inferred from higher FGM concentrations, needs to be further explored to tease apart
what activities are contributing to poor welfare. Work, including in the form of riding, may add benefit by
alleviating boredom in addition to supporting good physical and metabolic fitness, but overwork with little
time to socialize undoubtedly has negative effects.
Line 477: the authors indicate that mahouts are ‘the ones who know best when problems occur’ – this should
be rephrased. They are perhaps in the best position to detect problems early, as they work regularly with
their elephants. But they may not know how to recognise problems, so they must have training in order to do
so, and need a good working relationship with their elephant. This should be rephrased, and also these other
considerations discussed.
Response to the comment: This now reads “Generally, daily health checks are provided by the mahouts,
the people closest to the elephants, and in the best position to detect problems early. Mahouts may not
always recognize specific conditions, however, so conducting health workshops is imperative, as is having a
good relationship with the elephant.”.
Lines 487-489: use the term ‘bi-yearly’ should be biennially. Also, mention ‘since 2010’ twice in that
sentence.
Response to the comment: This now reads “International training courses and workshops for veterinarians
and biologists have been conducted yearly to every other year since 2010 in collaboration with CMU and the
NEI (Thailand), University of Peradeniya (Sri Lanka), Royal Veterinary College and Zoological Society of
London (UK), and Smithsonian Conservation Biology Institute (USA), with the support of Faculty of
Veterinary Medicine, Kasetsart University, British Council and Brian Nixon Fund for Protection of
Elephants in Thailand.”.
Lines 508-509: the authors indicate that FGM concentrations were higher in elephants involved in more
interactive tourist activities. Please can the authors provide a bit more detail to give this context? What were
the differences in FGM levels (ranges and mean/median values?). And how many camps/types of activities
were assessed?
Response to the comment: The sentence “Needed is a thorough evaluation of management practices at
observation-only camps, given the surprising finding that FGM concentrations were higher than in elephants
involved in more interactive tourist activities.” has been deleted.
Lines 533-534: the authors mention allostatic load indices, but describe them as markers to assess morbidity
and mortality risks – this is not really accurate. This model was developed to better understand and describe
the cumulative impacts of stress, particularly chronic stress, on an organism. This needs further explication
by the authors.
Response to the comment: This now reads “Another approach that has recently been applied to wildlife is
use of allostatic load indexes, a method developed to better understand and describe the cumulative impacts
of stress, particularly chronic stress, on an organism, and which can be been used to advise on morbidity and
mortality risks [Edwards et al., 2019].”.
Lines 539-555: the authors discuss the training methods of calves in Thailand, in the context of considering
the welfare impacts of types of training techniques used with the calves. However, the other very significant
feature of this way of training calves is the early separation of the calves from their mothers. This is in stark
contrast to natural behaviours in wild elephants, as females often remain in their natal group their entire
lives, and bulls only leave around 10-12 years of age when they reach puberty. This has profound welfare
implications for these calves that are separated from their mothers at an early age, and should also be
acknowledged and discussed.
Response to the comment: We have added “In Myanmar, mortality risk of calves is highest during the first
2 years of life, followed by another rise between 4-5 years of age when calves are weaned and traditional
training begins [Mar et al., 2012]. This situation is in contrast to natural behaviors in wild elephants,
particularly with regards to calves, where female offspring remain with their natal herd for life and males
emigrate at puberty [Sukumar, 1989]. While elephant behavior is usually discussed in workshops, more
training by animal behaviorists is needed to emphasize the importance of understanding individual elephant
behavior in the context of welfare.”.
Reference:
Sukumar, R., The Asian Elephant: Ecology and Management; Cambridge University Press: Cambridge,
1989.
Line 575: the authors suggest that it will be decades before changes are seen in elephants as a result of the
selective breeding that has taken place in elephant camps in Thailand over the past 10 years. This is not a
correct interpretation of the process of domestication, as this involves selective breeding over many
generations. Given the lifespan of elephants, this impact would likely take hundreds of years to be achieved,
so could only be realistically assessed after that time period. This needs to be corrected in the manuscript.
Response to the comment: We changed decades to hundreds of years. This now reads “It could be
hundreds of years before significant changes in temperament are observed given the 20+ year generation
interval”.

This manuscript is a resubmission of an earlier submission. The following is a list of the peer review reports and author responses from that submission.
Round 1
Reviewer 1 Report
Overall there is a lot of bias inherent in the way the manuscript was written, and not enough citations provided for statements which often sound more like opinion than citation of established facts.
"Thailand is the epicenter of elephant tourism"
-- suggest perhaps a different phrasing, 'epicenter' is not particularly scientific nor precise.
"the welfare of these elephants has been questioned in recent years by a number of animal activist groups" ; welfare concerns have also been raised by international tourism operators and many tourists, as well - these should also be mentioned, as this sentence makes it sound as though the criticisms are solely from animal activists, which is not correct.
"conclusions that the vast majority of captive elephants are abused are based on empirical evidence and not solid science"
this needs to be rephrased, as empirical evidence can be scientific.
line 26: "and fact with regards to what practices" - please correct grammar
line 27: "This paper aims to:" - please rephrase, a paper cannot aim to do anything. suggest 'the aim of this paper is...'
line 43: "Nowadays, Thailand is the epicenter of elephant tourism"
again, this is neither objective/scientific, nor particularly accurate. please rephrase
line 47: "Criticisms of venues that include tourist interactions with elephants, even touching"
'even touching' is unclear in this context; suggest rephrasing, perhaps 'and concern has even been raised over any type of physical contact with tourists' or words to that effect
lines 60-64 "Three organizations work cooperatively to register, census, microchip, and DNA fingerprint all captive elephants: the Department of Livestock Development (DLD), Ministry of Agriculture and Cooperatives; the Department of Provincial Administration, Ministry of Interior; and the Department of National Park, Wildlife and Plant Conservation, Ministry of Natural Resources."
It is unclear when this legislation came into effect, would be helpful to state a date from which this has taken place.
There is mention that a law was enacted in 1921 prohibiting capture of elephants from the wild, yet in 2017 a law was enacted to DNA fingerprint captive elephants, at least in part to prevent illegal capture of elephants from the wild. It would therefore be helpful to mention/discuss that although there was legislation as early as 1921 prohibiting wild capture, clearly it was happening, since there was obviously a law needed in 2017 to start to track this activity and address this problem. This should be discussed. It would also be helpful to discuss how well captive elephants in Thailand breed, and how self-sustaining the captive population is (to understand the context and potential drive for the capture of wild elephants).
lines 67-68:"Unregistered elephants not in the microchip or DNA databases are confiscated and sent to a government facility, usually the National Elephant Institute (NEI) in Lampang."
It would be helpful to discuss what the fate of these elephants is - what happens to them after they are sent to NEI? Are they released back into the wild? If not, how is their welfare ensured? What are the conditions under which they are kept?
The authors suggest that guidelines such as those used in western zoos should be implemented; while this is indeed a reasonable suggestion, as the authors have also highlighted, such guidelines are only as good as their enforcement, and this should also be discussed. Moreover, the limitations/deficiencies of those guidelines (for western or eastern zoos) with respect to providing good welfare for captive elephants should also be acknowledged, as it is clear from numerous welfare indicators that many elephants in western (and eastern) zoos also suffer from poor welfare.
line 110: "unfounded claims of universal brutality of elephants in tourism."
suggest rephrasing - right now it sounds like the elephants are committing brutalities.
Lines 117-138: the paragraph beginning with “Decreases in mahout quantity and quality are major concerns for elephant camps today,” makes many statements for which no evidence (citations) is provided, and the paragraph appears to have been written with an inherent bias, and numerous assumptions. Please can the authors rewrite and rephrase this paragraph. It would be more appropriate to begin the paragraph by discussing the concerns regarding the apparent lower status of the mahout profession in Thailand now compared to in the past, and as a result what concerns or issues there are for the mahout profession, and how this may well have an impact on the well being of the mahout, on the relationship between the mahout and his/her elephant, and how the elephant is treated, and thus the welfare of the elephant.
lines 143-145: "In answer to these questions, animal rights groups have created clear messages that use of an ankus and riding of elephants is always cruel, all elephants used for riding or shows have been brutally trained, and venues offering those activities should be boycotted." This concern has been raised by many different people and organisations, not just animal rights groups, and not just in connection with elephant handling in Thailand, but also in zoos. Please can the authors correct these statments and they also need to provide appropriate citations.
lines 155-156: "....where tourists have contact with an elephant without a barrier in between." the authors need to clarify this statment - presumably they mean a barrier between the tourists and the elephants.
lines 159-160: "campaigns are having a notable effect on Thailand’s tourism industry. Many camps now avoid riding, and hide the use of hooks and chains during working hours to satisfy tourists."
This needs to be rephrased, what needs to be made clear is that rather than comply with the stated wishes of the international community and stop using hooks and chains, operators are simply hiding their use from the tourists.
lines 160-162: "The number of overweight and obese elephants is growing due to reduced exercise opportunities, and because in many camps the only way for tourists to interact is through feeding (e.g., bananas, sugar cane)"
This statement assumes that the only way that captive elephants could get any exercise is by giving rides to tourists, but this is not correct. The authors should discuss that a problem perhaps of equal importance is that at these facilities, the owners/managers don't provide the elephants with any other opportunities for exercise. the elephants are often not allowed loose to interact freely with each other or to move around in their environment. If they were give large natural spaces to roam over, opportunity to swim/bathe/mud wallow and to forage naturally, this would provide both much better opportunities for exercise, but also for more natural activities and social interactions, which would be better for both physical and psychological health.
The authors refer to bathing and walking activities as ‘more intimate experiences’; this is a subjective interpretation, please can the authors rephrase.
The authors refer to glucocorticoids as ‘stress hormones’ – this is not correct, these hormones simply indicate arousal, and are also increased during times of physical activity, reproductive activity, during feeding, and in a circadian daily pattern (as the authors discuss later in the paper). Please can the authors correct this.
It is unclear what the point of Figure 1 is; it seems to be some kind of thought map, and is not appropriate for a peer reviewed paper.
lines 164-166: "As these changes take hold, one problem is how to ensure the safety of tourists, mahouts, and property if tools are not allowed. What are the alternatives? Mahout injuries and fatalities are a concern if they do not have access to the tools needed to safely control elephants." This paragraph discussing the welfare impacts on changes in training and management methods is inherently very biased, and does not consider methods such as positive reinforcement training and protected contact. While it is certainly true that it is essential for mahouts and others working with elephants to be able to do so safely, it is also the case that many elephants in Thailand and elsewhere have been injured (sometimes quite severely) through physical punishment with the ankus. This needs to be acknowledged and discussed by the authors sooner in the paper (it is discussed later, but the introduction to the topic in this paragraph is not balanced and has inherent bias).
typo in line 239 "forth" should be 'fourth'
lines 257-259: "So we conclude that, for the most part, camps in northern Thailand are doing a reasonably good job of providing adequate nutrition and water, which is reflected in good, if not sometimes excessive, body condition."
It is unclear if this conclusion was made based on the authors' observations of the feeding practices used by these camps, or if it was simply based on a survey asking for information from the camps, and the authors' conclusions were simply based on the answers given by the elephant camp owners/managers. If the latter, then the authors need to make this clear, and need to consider and discuss that there can be a large difference between answers provided by people (which may be subjective impressions, or they may just give the answers that they think the researchers want to hear), and the practices in which they actually engage.
Reviewer 2 Report
General comments:
I do think this paper is extremely important and I am glad there is more focus on captive elephant welfare in Thailand. My biggest concern about the paper though, is that it was supposed to be a review of the empirical literature only, but it seems that personal opinion was also included (e.g. statements such as, "Simply eliminating the ankus is probably not the panacea for elephant welfare that it is purported to be"; "Ultimately, we need to determine if 'no hook', 'no riding' policies really are all we need to ensure good welfare - we suspect not), which was not backed up by literature supporting this, but backed up by tradition. I am not aware of any papers investigating the use of bullhooks vs. no bullhooks on elephant welfare, and I think it is important to give this issue more emphasis in the review and call for more research objectively identifying the results of these two methods of training. One paper that may help that just recently came out in the past couple of years is:
Wilson, M. L., Perdue, B. M., Bloomsmith, M. A., & Maple, T. L. (2015). Rates of reinforcement and measures of compliance in free and protected contact elephant management systems. Zoo Biology, 34, 431 - 437.
I also think it is great that you mentioned that workshops and experts in the field are needed to address this issue, but I also think mentioning the need for behaviorists to also run workshops is also important.
Other than my comments above, I do think that the authors did a good job remaining reviewing other relevant literature and not making any grand claims about the findings of the literature.
Corrections to spelling/minor corrections:
Line 107: need to insert "in" between elephant managers and Thailand
Line 239: "forth" should be "fourth"
Line 476: "effects" should be "affects"
Reviewer 3 Report
General comments
This review provides a detailed, and well-written, account of the history of captive elephants in Thailand, as well as the current state of these animals (welfare, legal protection, etc.). There are a few sentences which contained more opinion and biased-argument, rather than objective statements supported by scientific references, but in general the authors kept the narrative a evidence-based as possible for this under-studied topic.
Specific comments:
Line 25: Do the authors mean “anecdotal evidence” rather than “empirical”?
Lines 103-104: The word “mahout” should be defined, as this is not a generally-known term. This word is first mentioned on line 76.
Lines 118-119: What ethnic group are the Mahouts coming from and where is the citation for this statement. There should also be a citation to support the claim that there is an issue with communication.
Lines 140-143: Is there evidence that these questions come from the public? Where is a citation for these questions?
Page 5 (no line numbers): “Authors note the use of nails hidden in pockets and mahout bags, which may be even more dangerous because the mahout needs be closer to correct an elephant than if an ankus, which extends the arm’s reach, was used.” Where is the citation for this statement? Do the authors have evidence that all mahouts use nails as an alternative? And the authors should define/describe what an “ankus” is.
Page 7 (no line numbers): “While some elephants with more gentle temperaments likely can be handled without an ankus, others cannot and mahouts and camp owners must know the difference and manage accordingly. Simply eliminating the ankus is probably not the panacea for elephant welfare that it is purported to be.” These statements are opinions and are not supported by the literature which was previously reviewed. The authors should be cautious in making emotional, un-supported claims like this in a scientific paper.